


**Assessment of relative importance of debris flow disaster risk affecting factors**
**based on meta-analysis — cases study of northwest and southwest China**
**Yuzheng Wang[1], Lei Nie[1], Min Zhang[1], Hong Wang[1], Yan Xu[1], and Tianyu Zuo[1]**
[1] College of Construction Engineering, Jilin University, Changchun 130026, Jilin, China
Correspondence: Min Zhang (minzhang@jlu.edu.cn)
**Abstract.** Debris flow is a type of special torrent containing numerous solid materials. It is
characterized by sudden outbreak, short duration, and strong destructive force. The occurrence of
debris flow is often affected by hydrogeological and geological conditions, including basin area, main
ditch length, relative height difference, slope, bed bending coefficient, daily maximum rainfall and so
on. With many types of factors affecting debris flow, no reliable basis for selecting factors to evaluate
debris flow risk has been established. Therefore, to study the factors affecting debris flow, exploring a
reliable method for assessing the relative importance of such factors is an important endeavor in debris
flow prevention and control work. In this research, debris flow risk assessment was combined with
meta-analysis to analyze quantitatively the relative importance of risk factors of debris flow in
northwest and southwest China. Results show that debris flow in northwest China is mainly affected by
topography and geological structure. Rainfall plays an important role in stimulating debris flow in this
area. For debris flow in southwest China, topography, geological structure, and rainfall conditions all
have considerable influence. Meta-analysis can provide a basis for the selection of risk factors of debris
flow and has certain reliability.
**Keywords:** debris flow, risk-affecting factors, relative importance, meta-analysis
**1 Introduction**
Debris flow is a type of sudden natural disaster in mountainous areas and a complicated natural
geographical process of landmarks. Debris flow disasters in the world have caused serious
infrastructure damage and casualties for centuries(Yu et al., 2018). Such disasters include the debris
flow hazards in eastern Philippines in 2006, which led to more than 300 houses buried and almost an
entire village of more than 1800 people killed, as well as the 2010 flooding and landslide disaster in
northeastern Brazil, where at least 44 people were killed and more than 1000 people went missing.
Debris flow also costs China up to 2 billion yuan a year in direct economic losses (Cui P et al., 2000).
Various environmental background factors affect the occurrence, development, movement,
accumulation, intensity, energy, and destructive power of debris flow, which has more than 70
kinds(Liu ,1996). An in-depth understanding and assessment of the risks of natural hazards is necessary
in order to develop sustainable risk management strategies including efficient damage mitigation
approaches(Kreibich et al., 2015; Kreibich et al., 2019). Hence, the comprehensive determination of
debris flow risk should not only consider scientific and correct factors but also such assessment's
comprehensiveness, representativeness, simplicity, and practicability.

The analysis and selection of the main impact factors of debris flow disaster and the study on the
impact of these factors on debris flow risk are conducive to the exploration of the main causes of debris
flow formation as well as lead to more a reasonable and targeted prevention and control of debris flow.



In existing studies, scholars selected different influencing factors for their respective research objects.
When Jiang Zhongxin (Jiang, 1992) established a simple discrimination method for debris flow gulch,
he selected the average of 24 h rainfall over many years, the storage of loose matter in the basin area,
lithology, and other influencing factors. To analyze the relationship between environmental factors and
landslides and debris flow disasters nationwide, (Zhang et al., 2009) selected six factors, including
elevation, elevation difference, slope, slope direction, vegetation type, and vegetation coverage. On the
basis of the "2 major factors plus 14 minor factors" proposed by Liu Xilin, (Chen et al., 2013) selected
the maximum outflow quantity and frequency of debris flow as major factors through a preliminary
screening of scatter diagram and the continued screening of rank correlation coefficient. Then, they
evaluated the risk of debris flow using seven minor factors, including the length of the main ditch.
Although some methods performed better than others, no single method proved to be superior in all
conditions(Reichenbach et al., 2018). According to the results of previous studies, the selection of
debris flow impact factors can be generally divided into single-channel study and regional study, and
the selection of impact factors has its own emphasis depending on the research environment.
Owing to the randomness of the determination of risk factors in debris flow assessment, the use of
meta-analysis to select debris flow risk assessment factors can provide a reliable basis for determining
these assessment factors. Meta-analysis refers to a scientific clinical research activity in which all
relevant studies are collected and rigorously evaluated and analyzed. In recent years, the research field
has been applied to various areas, including clinical medicine (Chandrasekaran et al., 2016; Schuetz et
al., 2018; Temple et al., 2018), ecology (Abdelraheem et al., 2017; Brustolin et al., 2018;
Chandrasekaran et al., 2016; Hedges, 1999; Lajeunesse, 2016; Li et al., 2018; Ma and Chen, 2016; Xu
and Yuan, 2017; Zhou et al., 2016), computer systems (Hong et al., 2018), and environmental and
energy applications (Marttunen et al., 2018). Therefore, the application of meta-analysis across
domains is imperative.
The remainder of this paper is organized as follows. Section 2 introduces the research question and the
six related debris flow risk factors. It also presents the selection, collection, and analysis data of these
factors. Section 3 describes our research methods and how meta-analysis is realized in this study.
Section 4 presents the results, starting with general information about the selected cases, followed by
the analyses of the research questions. Section 5 discusses the practical relevance of the results and
presents recommendations on how to diminish the risk of biases. Section 6 concludes the article.
**2 Materials**
**2.1 Selection of risk factors for debris flow**
The formation and evolution of debris flow disasters are controlled by a variety of time-space factors.
Taking into account the formation conditions and characteristics of debris flow and the statistical
principle of meta-analysis, six influential factors with obvious digital characteristics and quantifiable
characteristics are selected from the influencing factors of debris flow. These factors include relative
elevation (m), maximum daily precipitation (mm), longitudinal slope (%),drainage area (km$^2$), main
ditch slope (°), and length of main channel (km).
**Relative elevation (m):** This factor determines whether the loose material on the slope surface can be





activated to provide potential energy conditions for debris flow.

**Maximum daily precipitation (mm):** Continuous rainfall and heavy rain, especially extremely heavy
rainfall, are conducive to the stimulation of debris flow. The vast majority of debris flow is triggered by
(extraordinary) precipitation events(Bogaard and Greco, 2016). Slope softening caused by continuous
heavy rainfall will reduce the critical rainfall for debris flow initiation. The process of rainfall and
confluence carries with it a large amount of soil and rock, which then produce debris flow.

**Longitudinal slope (%):** The larger the longitudinal slope of the gully bed, the more rapid and
concentrated the high-speed water flow that will be formed in the process of precipitation in a short
period of time. Such water flow enhances the ability of water binding and erosion and can form debris
flow in a short period of time. Too large a gradient can also weaken the stability of surface material.

**Drainage area (km$^2$):** This factor reflects the status of sediment yield and confluence in the basin. The
accumulation of loose solid matter in the basin is affected by sediment yield, and the outbreak of debris
flow is closely related to the abundance of loose matter.

**Main ditch slope (°):** This factor has a controlling effect on the stress distribution in the slope, the
packing thickness of loose materials on the slope, and the thickness of vegetation. The larger the slope,
the greater the potential energy provided by the loose material source deposits, which weaken the
stability of the slope.

**Length of main ditch (km):** This factor reflects the flow distance of debris flow and the ability to
accept loose deposits along the way. Moreover, the damage to the downstream and gully can be judged
according to the gully length.

In addition to the above six factors, other geological factors, such as regional lithology, structure, and
weathering, and other economic factors, such as local grazing methods and human activities, also have
an important impact on the occurrence of disasters. However, given the statistical principle of
meta-analysis, the above six indicators are difficult to quantify and are thus selected as the impact
factors of debris flow.
**2.2 Data collection**
The data of debris flow risk assessment were collected by consulting the literature and reports on debris
flow disaster and risk assessment published in Chinese and in English in the last 10 years. Data
published in English were collected from the ISI-Web of Science (http://apps.webof knowledge.com/),
while data published in Chinese were collected from the China National Knowledge Infrastructure
(http://www. cnki.net/).

A total of 156 studies were retrieved, from which 93 that met the inclusion criteria were selected
through reading abstracts and titles, as well as the full text if necessary, and 63 were excluded. Among
the excluded literature, 17 were repeatedly published, and 46 were not consistent with the study
subjects or interventions. With the use of bibliometrics, the publication year, publication distribution,
and literature quality (methodology and experimental design) of the included studies were analyzed.

In terms of innovation theory, 26 out of the 93 references mentioned GIS support and APH model.
There were 22 articles related to grey relational degree and fuzzy judgment, 10 references to
geomorphological information entropy, 14 applications of extension method, and 21 references to
analytic hierarchy process and weight analysis.

From the aspect of research level, 39 of the 93 studies were about engineering technology and 54 about
basic and applied basic research. , and the less relevant technical guidance, advanced science and
technology, and standards and quality control are excluded.
**2.3 Data analysis**
Owing to the obvious differences in geological conditions and geological structures of debris flow
development in different regions, these two factors cannot be included in the meta-analysis index with
specific data. The study areas were grouped into two geographic regions: northwest China and
southwest China.
1.   Northwest China: This zone includes Inner Mongolia, Gansu, Xinjiang, Ningxia, and Shaanxi
provinces. Large and extra-large debris flows, which have the characteristics of wide distribution,
large scale, and heavy disaster, are mainly distributed in this area.

2.   Southwest China: This zone includes Guizhou, Yunnan, Chongqing, and Sichuan provinces.
Debris flows in this region are widely distributed, frequently active, and seriously harmful.

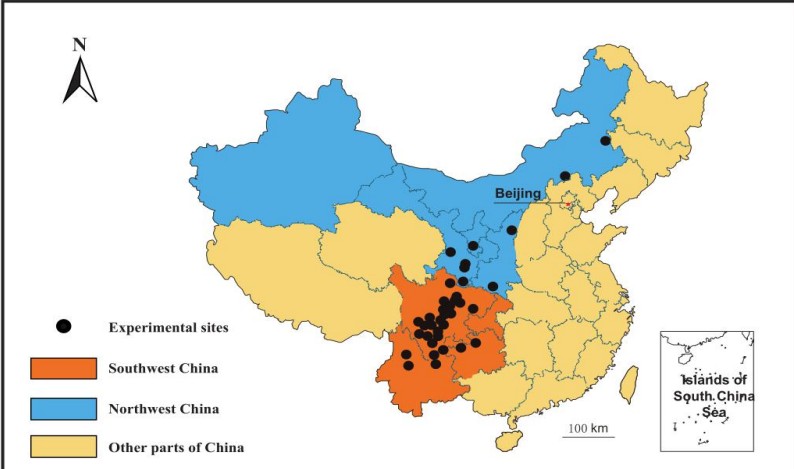


**Figure 1. Locations of debris flow disasters in the literature included in this meta-analysis.**

Standardized mean difference (SMD) of the two groups (experimental group and control group)
estimate the mean difference divided by the average standard deviation according to the landslide area,
which is divided into northwest and southwest. These areas each have 15 groups. Among them, the
northwest Tianjiagou debris flow and the 10 other debris flows are treated collectively as the control
group to calculate the maximum precipitation (mm), relative elevation difference (m), longitudinal
slope (%), basin area ($km^2$), long slope (°), and groove (km) as well as the other six factors affecting




the expectations and standard deviation. The data are shown in Table 1.

The corresponding indexes of other experimental groups were calculated, with 10 debris flows, such as
the Shuiqinggou debris flow, taken as examples as shown in Table 2.

In southwest China, 10 debris flows, including the Shenjiagou debris flow in Luding County, Sichuan
Province, were taken collectively as the control group. The expectation and standard deviation of six
influencing factors were calculated in the list, as shown in Table 3.

The corresponding indexes of other experimental groups were calculated, and 10 debris flows, such as
that in Ziluogou, Daocheng County, were taken as examples as shown in Table 4.
**3 Methods**
Meta-analysis is a scientific clinical research activity that refers to the comprehensive collection of all
relevant studies and their rigorous evaluation and analysis. It uses the quantitative synthesis method for
the statistical processing of data. Meta-analysis data can be divided into binary data and continuous
data. The influencing factors of debris flow to be studied in this research can be regarded as continuous
outcomes, also known as numerical variables.

For continuous variables, weighted mean difference and SMD are two important measures of SMD in
meta-analysis. In this study, due to the different dimensionality of relative height difference, daily
maximum precipitation, and other influencing factors, dimensional influence must be eliminated in the
analysis. In the effect index, SMD is obtained by dividing the estimated mean difference between the
two groups by the mean standard deviation. When the dimensional effects are eliminated, the results
can be combined. In SMD calculation, the expectation, standard deviation, and sample size of the
original study must be identified first. The weight of the mean difference of each original study is
determined by the accuracy of its effect estimation and is generally determined by variance or standard
deviation. SMD is a relative indicator that is unaffected by baseline risk and has good consistency.
Therefore, SMD was used as the effect indicator in this study.

Forest map, the most commonly used form of result expression in meta-analysis, was adopted in this
study. This method is based on statistical effect size and statistical analysis method (confidence
interval). In the statistical range, confidence interval refers to the distribution range of the real
measured values, which can reflect the accuracy of the results. In this meta-analysis, the Cochrane
systematic evaluation adopted the confidence interval range of 95%. In an ideal state, the objects
included in the meta-analysis should be absolutely homogeneous. However, due to the differences in
researchers, subjects, conditions, and other factors, the heterogeneity between studies "absolutely"
exists, so heterogeneity test is still needed. Meta-analysis of the Q statistic test and the $I^2$ test two
methods, the two indicators can be read at the bottom of the forest figure. The parameters are as
follows:

$$\text{Heterogeneity: Ta}u^2 = 0.00, Chi^2 = 27.89, df = 29(P = 0.52), I^2 = 0\%$$


Among the parameters, the first four are Q statistic test parameters, and the last item is on the test
parameters for $I^2$. In the Q statistic test, the P value (P > 0.1) was mainly used, so there was no


heterogeneity. Heterogeneity exists if P < 0.1.

In the inspection, the $I^2$ value was from 0 to 100%. According to the Cochrane handbook, if $I^2 \leq$
50%, then no heterogeneity exists; otherwise, heterogeneity exists.
**4 Results**
**4.1 Overview of the dataset**
Our dataset covers a total of 183 debris flow gullies evaluated by 47 authors in northwest China
and 158 debris flow gullies evaluated by 48 authors in southwest China. The two regions are studied
separately because the geomorphic and water source conditions of southwest and northwest China are
quite different. Each region was divided into a control group and 14 experimental groups according to
the similarity of geomorphic and water source conditions in the debris flow gully. After calculation, the
expected value and standard deviation of different debris flow groups in the two regions were obtained,
as shown in Tables 5 and 6, respectively.
**4.2 Influence of relative elevation on risk of debris flow**

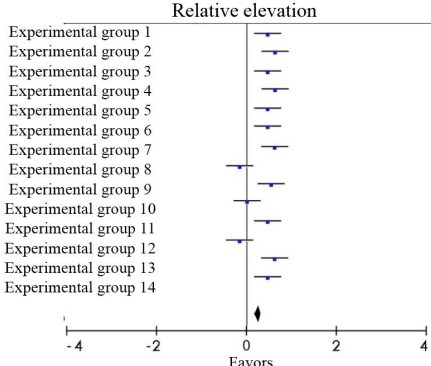


**Figure 2. Forest figure of the influence of relative elevation on debris flow in northwest China.**

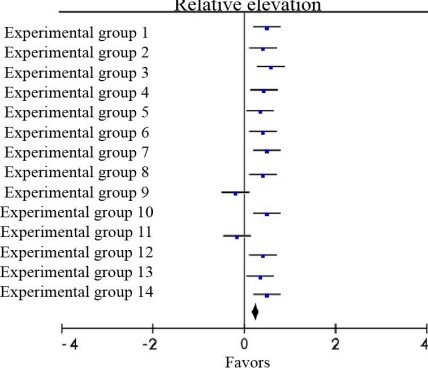


**Figure 3. Forest figure of the influence of relative elevation on debris flow in southwest China.**

Relative height difference determines whether the loose material on the slope surface can be activated





to provide the potential energy conditions needed for the generation of debris flow. Data of relative
height difference of debris flow in northwest and southwest China were selected to study the influence
degree of relative height difference on debris flow risk, including 14 cases in the experimental group
and 1 case in the control group. The influence degree of this influencing factor after regrouping is
shown in Figs. 2 and 3. In the northwest region, P = 0.27 and $I^2$ = 19% in the northwest of the forest
map of relative height difference of debris flow. In the southwest region, P = 0.16 and $I^2$ = 33%.
Statistical heterogeneity was small. The meta-analysis results are shown in Table 7, which reveals a
statistically significant difference between the experimental group and the control group. The influence
degree of relative height difference on debris flow risk in northwest and southwest regions was
analyzed through a comparison of the number of data points on the right side of the invalid vertical line
in the forest map with the total number of experimental data points.

**4.3 Influence of daily maximum precipitation on the risk of debris flow**

Rainstorms and continuous rainfall, especially extremely heavy rainfall, are conducive to the
stimulation of debris flow. The critical rainfall at which debris flow starts will be reduced by the
softening of the slope caused by continuous heavy rainfall. In the process of rainfall and confluence,
the solid materials in the gully are continuously scoured and a large number of soil and rock bodies are
carried, thus generating debris flow. Rainfall is an important excitation condition for debris flow. Given
the availability and accuracy of rainfall data, maximum daily precipitation is selected as the evaluation
index.

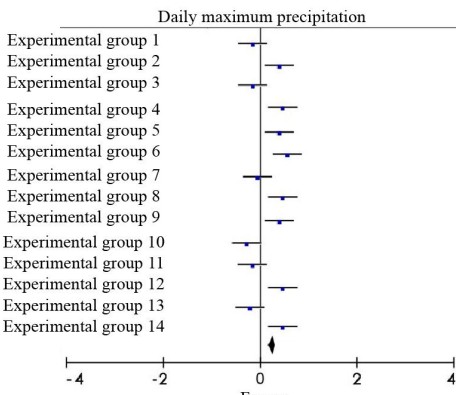


**Figure 4. Forest figure of the influence of daily maximum precipitation on debris flow in northwest China.**

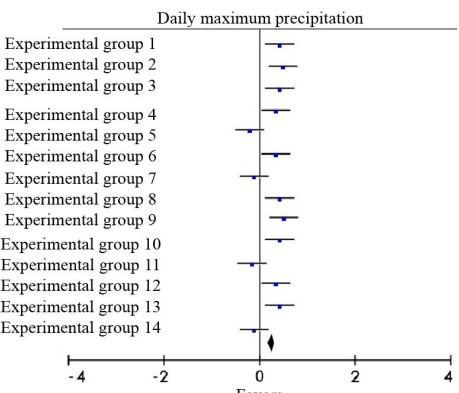

**Figure 5. Forest figure of the influence of daily maximum precipitation on debris flow in southwest China.**

Data of maximum daily rainfall of debris flow in northwest and southwest China were selected to study and compare the influence of maximum daily rainfall on debris flow occurrence indexes in northwest and southwest China, including 14 cases in the experimental group and 1 case in the control group. The influence degree of this influencing factor after regrouping is shown in Figs. 4 and 5. $P = 0.41$ and $I^2 = 4\%$ in the northwest of the forest map of the maximum daily precipitation of debris flow. In the southwest region, $P = 0.22$ and $I^2 = 9\%$. Statistical heterogeneity was small. The meta-analysis results are shown in Table 8, which reveals a statistically significant difference between the experimental group and the control group. Through a comparison of the number of data points on the right side of the invalid vertical line in the forest map with the total number of experimental data points, the influence degree of maximum precipitation on debris flow risk in northwest and southwest regions was analyzed.

**4.4 Influence of longitudinal slope of debris flow gully on risk of debris flow**

The larger the longitudinal slope of gully bed, the more rapid and concentrated the high-speed water flow that will be formed in the process of short-term concentrated precipitation, which strengthens water-binding ability and erosion and can form debris flow in a short time. Such water flow is the main factor of debris flow formation and movement. Moreover, too large a vertical slope will weaken the stability of surface materials and provide good source conditions for debris flow formation.




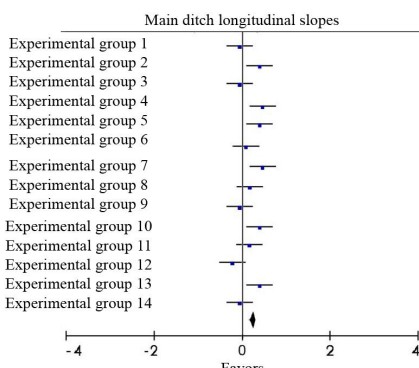


Figure 6. Forest figure of the influence of main ditch longitudinal slope on debris flow in northwest China.


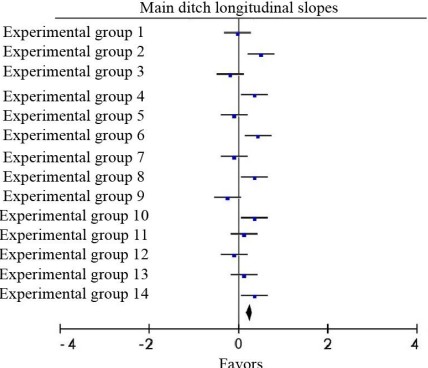


Figure 7. Forest figure of the influence of main ditch longitudinal slope on debris flow in southwest China.

Data of 15 groups of longitudinal slope of debris flow bed in northwest and southwest China were
selected to study and compare the influence degree of longitudinal slope of debris flow bed on debris
flow occurrence indexes in northwest and southwest China, including 14 cases in the experimental
group and 1 case in the control group. The influence degree of this influencing factor after regrouping
is shown in Figs. 6 and 7. In the northwest and southwest regions, P = 0.25 and $I^2$ = 4% and P = 0.35
and $I^2$ = 9%, respectively. Statistical heterogeneity was relatively high. As shown in Table 9,
statistically significant differences exist between the experimental group and the control group. The
influence degree of vertical slope of the main ditch on debris flow risk in the northwest and southwest
areas was analyzed through a comparison of the number of data points on the right side of the invalid
vertical line in the forest map with the total number of experimental data points.
**4.5 Influence of basin area on risk of debris flow**
The shape and size of the drainage basin have obvious influences on the process of rainfall and storm
runoff, which is directly related to the initiation and participation of loose debris in debris flow
activities. The influence factor to reflect the sediment and flow condition of the basin, the basin of
loose solid material accumulation quantity under the influence of sediment yield, and the outbreak of





debris flow is closely related to the rich loose material reserves.

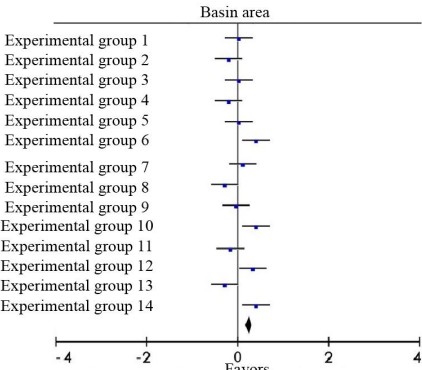


**Figure 8. Forest figure of the influence of basin area on debris flow in northwest China.**

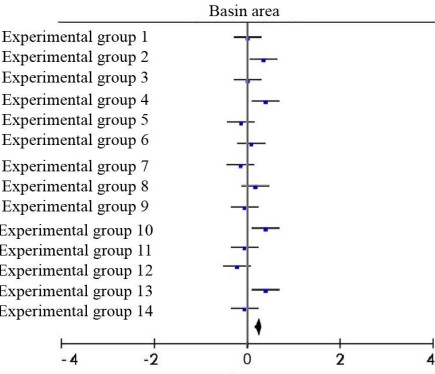


**Figure 9. Forest figure of the influence of basin area on debris flow in southwest China.**

A total 15 data research groups of debris flow basin area date in northwest and southwest China,
comprising 14 test groups and 1 control group, were selected to study the influence degree of the
northwest and southwest regional debris flow occurrence indicators. The influence degree of this
influencing factor after regrouping is shown in Figs. 8 and 9. In the northwest and southwest regions of
the forest area of debris flow basin, P = 0.36 and $I^2$ = 7%. In the southwest region, P = 0.35 and $I^2$ =
4%. Statistical heterogeneity was small. The results of meta-analysis are shown in Table 10, which
shows a statistically significant difference between the experimental group and the control group. The
influence degree of watershed area on debris flow risk in the northwest and southwest regions was
analyzed by comparing the number of data points on the right side of the invalid vertical line in the
forest map with the total number of experimental data points.
**4.6 Influence of main ditch slope on risk of debris flow**




Slope condition is the restriction condition of whether potential energy can be converted into kinetic
energy and conversion speed. Slope degree of ditch reflects the flatness of surface, which is the
potential factor of solid source material formation of debris flow. Slope plays a controlling role in stress
distribution, accumulation thickness of loose matter on the slope, and thickness of vegetation. The
larger the slope, the greater the potential energy provided by the loose material accumulation; and the
worse the stability of the slope, the greater the possibility of debris flow.

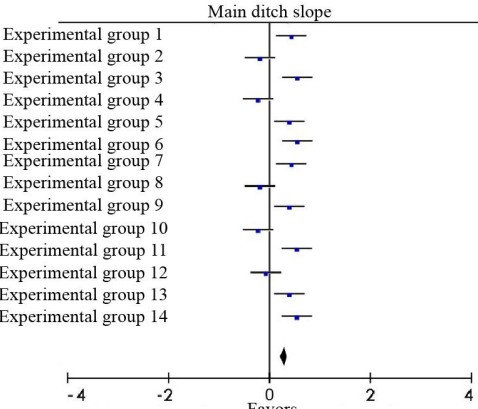


**Figure 10. Forest figure of the influence of main ditch slope on debris flow in northwest China.**

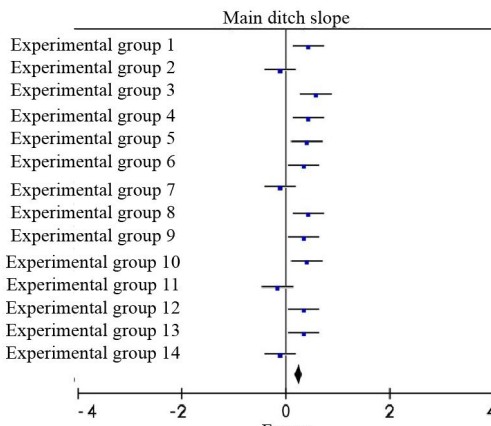


**Figure 11. Forest figure of the influence of main ditch slope on debris flow in southwest China.**

A total of 15 groups of debris flow slope data in northwest and southwest China, including 14 cases in
the experimental group and 1 case in the control group, were selected to study and compare the
influence degree of slope on debris flow occurrence indicators in northwest and southwest China. The
influence degree of this influencing factor after regrouping is shown in Figs. 10 and 11. In the
northwest region of the debris flow slope forest map, P = 0.34 and $I^2$ = 6%. In the southwest region, P
= 0.42 and $I^2$ = 19%. Statistical heterogeneity was small. The results of meta-analysis are shown in





Table 11, which reveals the statistically significant difference between the experimental group and the
control group. Through a comparison of the data points on the right of the invalid vertical line in the
forest map with the total number of experimental data points, the influence degree of slope on debris
flow risk in the northwest and southwest regions was analyzed.
**4.7 Influence of the length of main ditch on risk of debris flow**
The length of the main gully reflects the flow of debris flow and the ability to accept loose deposits
along the way. This length can be used as a basis for judging the destructive power of debris flow on
the downstream and gully mouth. It determines the flow of debris flow and how much loose solid
material is absorbed along the way. In addition, the farther the flow, the greater its energy and
destructive power will be.

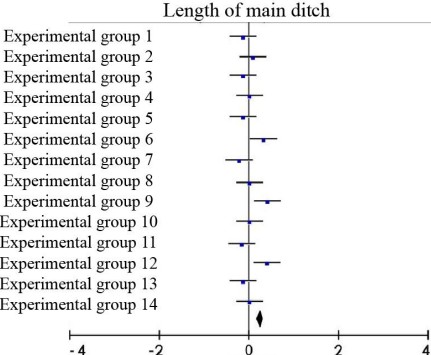


**Figure 12. Forest figure of the influence of length of main ditch on debris flow in northwest China.**

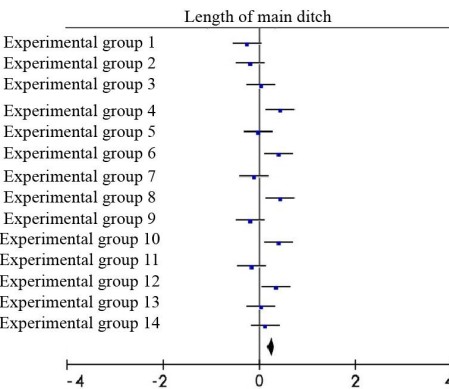


**Figure 13. Forest figure of the influence of length of main ditch on debris flow in southwest China.**

Data of the main gully length of debris flow in 15 groups were selected to study and compare the
influence degree of main gully length on debris flow occurrence indexes in northwest and southwest
regions, including 14 cases in the experimental group and 1 case in the control group. The influence
degree of this influencing factor after regrouping is shown in Figs. 12 and 13. In the figure of debris
flow gully length forest, $P = 0.42$ and $I^2 = 0\%$ in the northwest region and $P = 0.57$ and $I^2 = 10\%$ in




the southwest region. No statistical heterogeneity was found in the two regions. The results of
meta-analysis are shown in Table 12, which reveals a statistically significant difference between the
experimental group and the control group. Through a comparison of the number of data points on the
right side of the invalid vertical line in the forest map with the total number of experimental data points,
the influence of the length of the main gully on the risk of debris flow in the northwest and southwest
regions was analyzed.
**5 Discussion**
Through the above meta-analysis, the influences of various influencing factors on debris flow
excitation in southwest and northwest China are obtained, as shown in Tables 13 and 14.

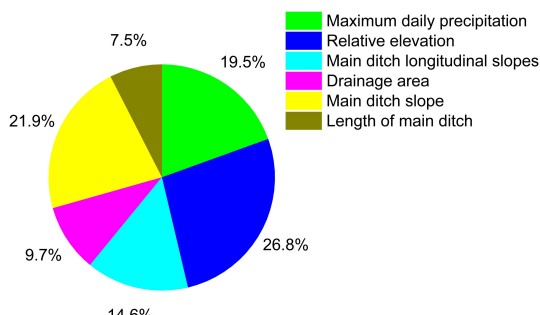


**Figure 14. Influence degree of debris flow factors on debris flow excitation in northwest China.**

According to the order of the above debris flow influencing factors based on their influence degree on
debris flow excitation in northwest China, the three factors with the highest influence degrees can be
obtained as follows: relative height difference, slope, and maximum daily precipitation. Among them,
relative elevation accounts for the largest proportion in the influence degree of all factors, up to 26.8%.
This finding indicates that topographic and tectonic factors play a major role in the occurrence and
spatial distribution of debris flows in northwest China, and maximum daily precipitation has a great
influence on the stimulation of debris flows. This result is attributed to the extensive distribution of
weak rocks in northwest China, including a large number of structural fault zones, the significant
influence of neotectonic movement, extremely developed fold faults, and poor integrity. The northwest
area is mountainous, and the new and old diluvial fans develop in the mountain pass, which provides
the source foundation for debris flow.



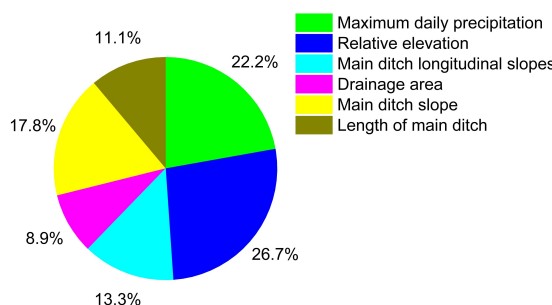


**Figure 15. Influence degree of debris flow factors on debris flow excitation in southwest China.**


Similarly, according to the influence degree of debris flow in southwest China on triggering debris flow, the influencing factors are ranked, and the three factors with higher influence degree are relative height difference, maximum rainfall and slope. It can be seen that the topographic, geological and structural factors and daily maximum precipitation in southwestern China play a dominant role in the occurrence and spatial distribution of debris flows. Among them, relative elevation accounts for the largest proportion in the influence degree of all factors, up to 26.7%. Maximum daily precipitation has greater impact on debris flow in southwest China than in Northwest China. This result is due to the complex terrain in southwest China, which includes five geomorphic units, including plateau, plain, mountain, hill, and basin. Therefore, the range of elevation variation is large, and the huge fluctuation of the terrain makes for an unstable geomorphic structure, providing a certain potential energy for debris flow materials and laying a foundation for the occurrence of geological disasters. Steep slopes and the availability of loose debris in these areas provide suitable topographic conditions and source materials for debris flows (Liu et al., 2016). Southwest China has a special climate with significant regional differences in performance. The climate varies greatly vertically, the dry rainy season is distinct, and the summer rainfall is concentrated and heavy. If the vegetation cover is not good, then the loose debris material on the hillside will cause soil and water loss under the erosion of precipitation and runoff. Therefore, the maximum precipitation in the southwest region has a greater impact on debris flow excitation than that in the northwest region.

A certain difference can be observed between the results obtained from meta-analysis and the current widely recognized "2 major factors plus 14 minor factors" method proposed by Liu Xilin in domestic industry. These results are mainly affected by human factors, such as the selection of sample and sample area. To reduce this error, the following improvements can be made:

1. When selecting research samples, try to select samples from areas with similar geological environments or similar geographical locations to the area for evaluation.
2. In the selection of evaluation factors, risk factors with the characteristics of the region must be first removed, then the risk factors with more universal, quantifiable, and obvious digital characteristics can be selected.



3. When the effects of several risk factors are roughly equal, meta-analysis can be conducted for
these risk factors after sample expansion.

## 6. Conclusions

With debris flow in China taken as an example, this study collected and collated a large number of data
of debris flow. It also selected six factors from various factors affecting debris flow for meta-analysis
and compared the results of the analysis. This study provides a reliable basis for the selection of debris
flow factors. The conclusions are as follows:
1. The feature of meta-analysis is that researchers synthetically analyze the results obtained from
previous studies to reflect regular patterns in a more objective form. It can provide a basis for
the selection of risk factors of debris flow and has certain reliability.
2. Debris flow in northwest China is mainly affected by the topography and geological structure.
Rainfall plays an important role in stimulating debris flow in this area. In southwest China,
topography, geological structure, and rainfall conditions have a great influence on debris flow.
Maximum daily precipitation has greater impact on debris flow in southwest China than in
northwest China.
3. Given that debris flow occurs in different regions, the selection of risk factors is closely related
to the region where the debris flow occurs. Samples from similar geological environments or
geographical locations should be selected for analysis when screening risk factors.

*Data availability.* The data are available from the authors upon request.
*Author contributions.* YW undertook the work and wrote them manuscript under the supervision of LN and MZ.
YX, HW and TZ helped with test data collection and numerical analysis.
*Competing interests.* The authors declare that they have no conflict of interest.
*Acknowledgements.* This work was supported by the National Science Foundation of China (Grant No.41572254
41502322, 41702300), the Science and technology development project of Jilin Province, China (Grant
No.20180520073JH), and the Jilin University Outstanding Youth Foundation.

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






**Table 1.** Influencing factors for the control group in northwest China

| Experimental group | Maximum daily precipitation (mm) | Relative elevation (m) | Main ditch longitudinal slopes (%) | Drainage area (km²) | Main ditch slope (°) | Length of main ditch (km) |
|---|---|---|---|---|---|---|
| Tianjiagou (1) | 84.7 | 369.7 | 205.5 | 1.1 | 26.8 | 1.9 |
| Tianjiagou (2) | 60.1 | 339.1 | 126.8 | 1.2 | 36.1 | 3.5 |
| Tianjiagou (3) | 99.5 | 433.7 | 121.1 | 1.8 | 31.5 | 5.6 |
| Huachi (1) | 69.0 | 329.1 | 174.6 | 1.4 | 31.4 | 2.9 |
| Huachi (2) | 92.1 | 428.9 | 141.6 | 1.7 | 39.7 | 4.6 |
| Honghegou (1) | 96.8 | 432.8 | 132.6 | 1.9 | 33.3 | 4.6 |
| Honghegou (2) | 81.3 | 449.4 | 196.7 | 1.2 | 39.4 | 5.5 |
| Meijiagou (1) | 101.1 | 393.3 | 194.8 | 1.6 | 29.2 | 2.1 |
| Meijiagou (2) | 100.2 | 337.2 | 149.0 | 1.7 | 34.1 | 3.4 |
| Meijiagou (3) | 58.8 | 443.6 | 149.2 | 1.3 | 32.3 | 2.6 |
| Value of expectation | 84.3 | 395.7 | 159.2 | 1.5 | 33.4 | 3.7 |
| Standard of deviation | 16.6 | 48.1 | 31.2 | 0.3 | 4.1 | 1.3 |


**Table 2.** Influencing factors for the experimental group in northwest China

| Experimental group | Maximum daily precipitation (mm) | Relative elevation (m) | Main ditch longitudinal slopes (%) | Drainage area (km²) | Main ditch slope (°) | Length of main ditch (km) |
|---|---|---|---|---|---|---|
| Shuijinggou (1) | 59.3 | 353.0 | 127.0 | 1.2 | 31.5 | 2.1 |
| Shuijinggou (2) | 78.6 | 340.1 | 131.3 | 1.8 | 20.8 | 5.0 |
| Shuijinggou (3) | 93.9 | 390.8 | 183.7 | 2.1 | 41.4 | 2.4 |
| Shangzhuogou (1) | 89.2 | 340.0 | 198.8 | 1.9 | 32.6 | 2.1 |
| Shangzhuogou (2) | 72.2 | 318.4 | 173.5 | 2.2 | 22.2 | 2.5 |
| Shangzhuogou (3) | 63.1 | 360.0 | 125.2 | 1.8 | 29.1 | 2.3 |
| Sanyanyugou (1) | 102.8 | 400.4 | 141.9 | 1.5 | 35.8 | 2.0 |
| Sanyanyugou (2) | 103.9 | 339.5 | 147.9 | 1.3 | 35.2 | 2.2 |
| Sanyanyugou (3) | 75.5 | 440.2 | 127.3 | 1.8 | 31.5 | 1.8 |
| Sanyanyugou (4) | 68.3 | 321.9 | 156.2 | 2.2 | 30.5 | 1.5 |
| Value of expectation | 80.7 | 360.4 | 151.3 | 1.8 | 31.1 | 2.4 |
| Standard of deviation | 16.0 | 38.7 | 26.2 | 0.4 | 6.1 | 1.0 |








**Table 3.** Influencing factors of control group in southwest China

| Experimental group | Maximum daily precipitation (mm) | Relative elevation (m) | Main ditch longitudinal slopes (%) | Drainage area (km²) | Main ditch slope (°) | Length of main ditch (km) |
|---|---|---|---|---|---|---|
| Shenjiagou (1) | 92.6 | 461.5 | 190.5 | 2.0 | 38.1 | 5.1 |
| Shenjiagou (2) | 106.2 | 331.2 | 216.7 | 1.5 | 20.2 | 2.5 |
| Shenjiagou (3) | 88.9 | 458.1 | 128.7 | 1.5 | 37.9 | 3.0 |
| Guandigou (1) | 73.4 | 414.6 | 222.6 | 1.8 | 37.2 | 5.4 |
| Guandigou (2) | 119.0 | 347.5 | 199.8 | 1.9 | 26.2 | 4.5 |
| Qinglinggou (1) | 92.3 | 437.1 | 238.4 | 1.3 | 29.9 | 3.7 |
| Qinglinggou (2) | 118.8 | 444.8 | 221.8 | 2.3 | 27.8 | 5.2 |
| Qinglinggou (3) | 118.1 | 469.4 | 150.2 | 1.7 | 25.2 | 2.3 |
| Yijiagou (1) | 70.6 | 335.9 | 189.4 | 1.6 | 31.2 | 5.2 |
| Yijiagou (2) | 93.3 | 361.1 | 164.3 | 1.4 | 22.9 | 3.0 |
| Value of expectation | 97.3 | 406.1 | 192.2 | 1.7 | 29.7 | 4.0 |
| Standard of deviation | 17.8 | 56.1 | 35.2 | 0.3 | 6.4 | 1.2 |


**Table 4.** Influencing factors for the experimental group in southwest China

| Experimental group | Maximum daily precipitation (mm) | Relative elevation (m) | Main ditch longitudinal slopes (%) | Drainage area (km²) | Main ditch slope (°) | Length of main ditch (km) |
|---|---|---|---|---|---|---|
| Ziluogou (1) | 106.3 | 408.9 | 195.9 | 2.1 | 38.7 | 2.8 |
| Ziluogou (2) | 118.3 | 380.3 | 195.5 | 1.3 | 45.0 | 3.7 |
| Ziluogou (3) | 84.2 | 361.4 | 194.6 | 2.0 | 44.1 | 1.7 |
| Dongxianggou (1) | 109.1 | 440.3 | 192.4 | 2.2 | 37.5 | 2.3 |
| Dongxianggou (2) | 79.7 | 385.5 | 236.1 | 2.2 | 43.2 | 3.3 |
| Laogangou (1) | 106.3 | 328.6 | 184.4 | 1.6 | 37.5 | 4.5 |
| Laogangou (2) | 116.7 | 389.1 | 141.7 | 1.3 | 29.0 | 5.3 |
| Shuzhenggou (1) | 97.3 | 331.2 | 192.5 | 2.2 | 36.2 | 5.1 |
| Shuzhenggou (2) | 105.8 | 353.7 | 121.9 | 2.5 | 41.0 | 5.1 |
| Shuzhenggou (3) | 105.8 | 464.3 | 179.0 | 2.1 | 36.8 | 3.7 |
| Value of expectation | 103.0 | 384.3 | 183.4 | 2.0 | 38.9 | 3.8 |
| Standard of deviation | 12.6 | 44.2 | 31.5 | 0.4 | 4.7 | 1.2 |








**Table 5.** Expected value (E) and standard deviation (SD) of risk factors in northwest China

| Group | Maximum daily precipitation (mm) | | Relative elevation (m) | | Main ditch longitudinal slopes (%) | | Drainage area (km²) | | Main ditch slope (°) | | Length of main ditch (km) | |
|---|---|---|---|---|---|---|---|---|---|---|---|---|
| | E | SD | E | SD | E | SD | E | SD | E | SD | E | SD |
| Experimental group 1 | 82.8 | 10.8 | 375.0 | 46.7 | 166.7 | 26.8 | 1.4 | 0.3 | 30.8 | 7.9 | 3.2 | 1.1 |
| Experimental group 2 | 74.4 | 12.7 | 396.0 | 40.4 | 166.0 | 26.1 | 1.7 | 0.3 | 33.9 | 8.3 | 4.5 | 0.9 |
| Experimental group 3 | 84.8 | 12.8 | 403.3 | 37.5 | 156.1 | 30.9 | 1.7 | 0.3 | 29.5 | 6.6 | 3.6 | 1.5 |
| Experimental group 4 | 80.7 | 16.0 | 360.4 | 38.7 | 151.3 | 26.2 | 1.8 | 0.3 | 31.1 | 6.1 | 2.4 | 1.0 |
| Experimental group 5 | 88.4 | 13.9 | 359.2 | 44.2 | 155.6 | 25.9 | 1.7 | 0.3 | 32.3 | 8.5 | 3.4 | 1.2 |
| Experimental group 6 | 77.6 | 14.0 | 388.7 | 42.0 | 155.2 | 29.9 | 1.8 | 0.3 | 28.8 | 5.5 | 4.1 | 1.3 |
| Experimental group 7 | 76.3 | 11.3 | 381.1 | 37.7 | 145.9 | 25.7 | 1.3 | 0.1 | 33.3 | 8.3 | 3.3 | 0.9 |
| Experimental group 8 | 86.4 | 10.9 | 361.6 | 52.8 | 154.0 | 23.5 | 1.6 | 0.3 | 32.1 | 7.7 | 3.4 | 1.0 |
| Experimental group 9 | 75.9 | 10.2 | 382.5 | 45.7 | 165.4 | 23.9 | 1.6 | 0.3 | 33.4 | 6.3 | 3.9 | 1.1 |
| Experimental group 10 | 89.6 | 14.3 | 355.5 | 33.2 | 150.1 | 30.4 | 1.8 | 0.3 | 28.5 | 7.7 | 3.6 | 1.5 |
| Experimental group 11 | 74.9 | 12.2 | 363.2 | 41.4 | 152.2 | 27.3 | 1.7 | 0.3 | 38.2 | 4.2 | 3.8 | 1.2 |
| Experimental group 12 | 85.5 | 12.0 | 365.1 | 37.8 | 152.8 | 32.1 | 1.6 | 0.4 | 31.3 | 6.9 | 3.8 | 1.4 |
| Experimental group 13 | 80.6 | 15.2 | 384.2 | 35.5 | 153.1 | 26.3 | 1.6 | 0.4 | 30.6 | 7.5 | 3.5 | 1.1 |
| Experimental group 14 | 82.0 | 15.0 | 392.1 | 37.5 | 171.2 | 32.1 | 1.6 | 0.3 | 34.0 | 6.9 | 3.5 | 1.1 |
| Control group | 84.3 | 16.6 | 395.7 | 48.1 | 159.2 | 31.2 | 1.5 | 0.3 | 33.4 | 4.1 | 3.7 | 1.3 |


**Table 6**. Expected value (E) and standard deviation (SD) of risk factors in southwest China

| Group | Maximum daily precipitation (mm) | | Relative elevation (m) | | Main ditch longitudinal slopes (%) | | Drainage area (km²) | | Main ditch slope (°) | | Length of main ditch (km) | |
|---|---|---|---|---|---|---|---|---|---|---|---|---|
| | E | SD | E | SD | E | SD | E | SD | E | SD | E | SD |
| Experimental group 1 | 96.5 | 14.1 | 417.8 | 51.4 | 164.4 | 38.7 | 1.9 | 0.3 | 31.3 | 7.6 | 3.7 | 1.0 |
| Experimental group 2 | 93.7 | 17.0 | 421.4 | 37.9 | 194.4 | 33.6 | 1.7 | 0.3 | 31.2 | 7.3 | 3.6 | 1.2 |
| Experimental group 3 | 100.8 | 19.2 | 398.8 | 38.4 | 174.9 | 35.3 | 1.8 | 0.3 | 34.6 | 7.3 | 3.0 | 1.0 |
| Experimental group 4 | 100.5 | 11.2 | 403.5 | 60.0 | 190.5 | 29.5 | 2.1 | 0.4 | 29.9 | 5.7 | 2.9 | 0.9 |
| Experimental group 5 | 99.3 | 16.9 | 384.9 | 40.4 | 204.4 | 40.3 | 1.7 | 0.2 | 28.5 | 5.5 | 3.6 | 1.1 |
| Experimental group 6 | 98.9 | 14.1 | 413.5 | 42.1 | 163.1 | 25.3 | 1.8 | 0.4 | 30.9 | 6.1 | 3.9 | 1.0 |
| Experimental group 7 | 102.9 | 12.6 | 384.3 | 44.2 | 183.4 | 31.5 | 1.9 | 0.4 | 38.9 | 4.7 | 3.7 | 1.3 |
| Experimental group 8 | 86.8 | 12.7 | 398.4 | 45.8 | 189.7 | 34.0 | 2.0 | 0.3 | 27.0 | 5.1 | 3.6 | 1.1 |
| Experimental group 9 | 95.8 | 18.5 | 397.2 | 40.4 | 186.9 | 33.9 | 2.0 | 0.3 | 31.1 | 6.0 | 3.0 | 1.2 |
| Experimental group 10 | 86.9 | 14.6 | 414.9 | 53.4 | 187.1 | 29.4 | 1.8 | 0.4 | 30.8 | 5.5 | 3.6 | 1.1 |
| Experimental group 11 | 86.1 | 20.7 | 427.6 | 31.6 | 169.4 | 45.2 | 1.6 | 0.3 | 33.0 | 6.8 | 3.4 | 1.3 |
| Experimental group 12 | 94.2 | 14.4 | 414.3 | 47.8 | 187.4 | 24.3 | 1.9 | 0.3 | 32.2 | 6.6 | 3.4 | 1.2 |
| Experimental group 13 | 96.6 | 14.6 | 405.2 | 45.6 | 182.8 | 31.9 | 1.8 | 0.4 | 32.8 | 6.4 | 3.1 | 1.0 |
| Experimental group 14 | 84.0 | 13.0 | 362.5 | 31.1 | 165.2 | 33.5 | 1.9 | 0.3 | 31.1 | 6.7 | 3.6 | 1.2 |
| Control group | 97.3 | 17.8 | 406.1 | 56.1 | 192.2 | 35.2 | 1.7 | 0.3 | 29.6 | 6.4 | 4.0 | 1.2 |





**Table 7**. Influence degree of relative elevation on risk of debris flow

| Group | P | $I^2$(%) | Z | Valid point | Total number |
|---|---|---|---|---|---|
| Northwest China | 0.27 | 19 | 2.92 | 11 | 14 |
| Southwest China | 0.16 | 33 | 2.6 | 12 | 14 |


**Table 8.** Influence degree of daily maximum precipitation on risk of debris flow

| Group | P | $I^2$(%) | Z | Valid point | Total number |
|---|---|---|---|---|---|
| Northwest China | 0.41 | 4 | 2.79 | 8 | 14 |
| Southwest China | 0.22 | 9 | 2.30 | 10 | 14 |


**Table 9.** Influence degree of main ditch longitudinal slope on risk of debris flow

| Group | P | $I^2$(%) | Z | Valid point | Total number |
|---|---|---|---|---|---|
| Northwest China | 0.25 | 4 | 2.53 | 6 | 14 |
| Southwest China | 0.35 | 9 | 2.38 | 6 | 14 |


**Table 10.** Influence degree of basin area on risk of debris flow

| Group | P | $I^2$(%) | Z | Valid point | Total number |
|---|---|---|---|---|---|
| Northwest China | 0.36 | 7 | 3.34 | 4 | 14 |
| Southwest China | 0.35 | 4 | 2.56 | 4 | 14 |


**Table 11.** Influence degree of main ditch slope on risk of debris flow

| Group | P | $I^2$(%) | Z | Valid point | Total number |
|---|---|---|---|---|---|
| Northwest China | 0.34 | 6 | 2.56 | 9 | 14 |
| Southwest China | 0.42 | 19 | 2.37 | 8 | 14 |


**Table 12**. Influence degree of length of main ditch on risk of debris flow

| Group | P | $I^2$(%) | Z | Valid point | Total number |
|---|---|---|---|---|---|
| Northwest China | 0.42 | 0 | 2.56 | 3 | 14 |
| Southwest China | 0.57 | 10 | 2.11 | 5 | 14 |


**Table 13.** Influence degree of debris flow factors on debris flow excitation in northwest China

| Group | Maximum daily precipitation (mm) | Relative elevation (m) | Main ditch longitudinal slopes (%) | Drainage area (km$^2$) | Main ditch slope (°) | Length of main ditch (km) |
|---|---|---|---|---|---|---|
| Proportion of influence degree (%) | 19.5 | 26.8 | 14.6 | 9.7 | 21.9 | 7.5 |






**Table 14.** Influence degree of debris flow factors on debris flow excitation in southwest China

| Group | Maximum daily precipitation (mm) | Relative elevation (m) | Main ditch longitudinal slopes (%) | Drainage area (km²) | Main ditch slope (°) | Length of main ditch (km) |
|---|---|---|---|---|---|---|
| Proportion of influence degree (%) | 22.2 | 26.7 | 13.3 | 8.9 | 17.8 | 11.1 |
