# Peer review of "Assessment of relative importance of debris flow disaster risk affecting factors"

_Natural Hazards and Earth System Sciences, 2019_

## Referee Comment (RC1) · Anonymous Referee #1 · 29 Jan 2020

I think the manuscript may be resubmitted to NHESS after re-writing. Instead of a scientific paper, presently the manuscript looks like a technical report short of enough analysis in-depth. Writing problems exist throughout the manuscript. For examples (not all): 1. The abstract looks like that of a review paper. No any quantitative result and in depth analysis is found. I also can not found any quantitative result in the section "6. Conclusions". 2. Figures 1-2 are strange and unreadable. Furthermore, they look very similar. Why do not you merge them after a major revision? Please check other figures, e.g., Figs. 3-13. 3. Tables 1-3 seem very similar. Instead of the original data,

might you sum up any common regularity from the three tables? Please check other tables.

---

## Referee Comment (RC2) · Anonymous Referee #2 · 30 Jan 2020

Authors should introduce the description of the occurrence mechanism of the analyzed debris flows: are they runoff-generated debris flows? Are they channelized debris flows? References about them should be also inserted (Coe et al., 2008; Gregoretti and Dalla Fontana, 2008; Hurlimann et al., 2014; Ma et al., 2018). Channelized debris flows have a very high impact on the environment because the grow volumetrically along routing (Reid e tal., 2016) with values also up to 100000 m3 (Gregoretti et al., 2018). The examined factors should depend on such analysis.

About the factors, authors should better distinguish between longitudinal slope and
main ditch slope. The former is the gully bed longitudinal slope, while the latter is the main slope of what? What do the authors mean for ditch? Moreover, looking at what it is written at lines 251 and 317, the two slopes seem to be the same thing. About factors the presence of sediment source area can play a significant role (see the discussion) and maybe the hourly precipitation is better than the daily precipitations.

Some sentences are unclear, with no apparent meaning and out of the context (see below some examples)

Lines 88-89: "The process of rainfall and confluence carries with it a large amount of soil and rock, which then produce debris flow" unclear sentence.

Lines 172-173: "For continuous variables, weighted mean difference and SMD are two important measures of SMD in meta-analysis." Unclear sentence

Lines 393-395 "The feature of meta-analysis is that researchers synthetically analyze the results obtained from previous studies to reflect regular patterns in a more objective form. It can provide a basis for the selection of risk factors of debris flow and has certain reliability." Unclear periods

The discussion of results seems poor.

The writer suggests the authors to re-write the paper, better explaining the phenomen, linking the factors to the physics of debris flow occurrence and widening the discussion of results.

Coe, J.A., Kinner, D.A., Godt, J.W., 2008. Initiation conditions for debris ïñĆows generated by runoff at Chalk Cliffs, central Colorado. Geomorphology 96, 270–297.

Gregoretti, C., Dalla Fontana, G., 2008. The triggering of debris ïñĆow due to channel-bed failure in some alpine headwater basins of the Dolomites: analyses of critical runoff. Hydrol. Process. 22, 2248–2263. https://doi.org/10.1002/hyp.6821.

Gregoretti, C., Degetto, M., Bernard, M., Boreggio, M., 2018. The debris ïñĆow occurred at Ru Secco Creek, Venetian Dolomites, on 4 August 2015: analysis of the phenomenon, its characteristics and reproduction by models. Front. Earth Sci. 6, 80. https://doi.org/10.3389/feart.2018.00080.

Hurlimann, M., Abanco, C., Moya, J., Vilajosana, I., 2014. Results and experiences gathered at the rebaixader debris-flow monitoring site, Central Pyrenees, Spain, Landslides. https://doi.org/10.1007/s10346-013-0452-y\ignorespaces161-175.

Kean, J.W., McCoy, S.W., Tucker, G.E., Staley, D.M., Coe, J.A., 2013. Runoff-generated debris flows: observations and modeling of surge initiation, magnitude and frequency. J. Geophys. Res. 118, 2190–2207. https://doi.org/10.1029/jgrf20148.

Ma, C., Deng, J., Wang, R., 2018. Analysis of the triggering conditions and erosion of a runoff triggered debris flow in Miyun County, Beijing, China. Landslide https://doi.org/10.1007/s10346-018-1080-3.

Reid, M. E., Coe, J. A., and Dianne, L. B. (2016). Forecasting inundation from debris flows that grows volumetrically during travel, with application to the Oregon Coast Range, USA. Geomorphology 273, 396–411. doi: 10.1016/j.geomorph.2016.07.039

---

## Referee Comment (RC3) · Anonymous Referee #3 · 4 Feb 2020

I have carefully read the paper and evaluated its potential contribution on the analysis of risk factors of debris flow. I regretted to point out that the paper appears to be a technical note on promoting existing and well-established scientific clinical research statistical tool, the "meta-analysis". It doesn't seem to be a research article for me. Here are the main points that I think the paper is far from the standards of publication in the NHESS journal.

1. As noted in abstract: "The occurrence of debris flow is often affected by hydro-geological and geological conditions, including basin area, main ditch length, relative

height difference, slope, bed bending coefficient, daily maximum rainfall..." However, only six parameters have been selected. 2. All six parameters can classified into two groups, geomorphologic and rainfall, and parts are too similar. Nevertheless, what are the effects of other factors, e.g. lithological and structural conditions, vegetation, human activity etc.... 3. Potential advantages of meta-analyses are clear, however, they also have the potential to mislead seriously, e.g. study designs, within-study biases, variation across studies, and reporting biases etc... Moreover, like any tool, statistical methods can be misused. The phenomena of selective outcome reporting and publication bias are likely occurred in this manuscript. 4. The tables and figures are not well prepared, and may able to merged and simplified 5. No major conclusion.

Thus, I regret to recommend rejection of the paper in its current form. A resubmission may be encouraged if the new improved paper will promise addressing explicitly.

---

## Author Comment (AC1) · 16 Feb 2020

I am very grateful to your comments for the manuscript. According with your advice, we amended the relevant part in manuscript. Some of your questions were answered below. Major comments: 1. The abstract looks like that of a review paper. No any quantitative result and in depth analysis is found. I also can not found any quantitative result in the section "6. Conclusions". 2. Figures 1-2 are strange and unreadable. Furthermore, they look very similar. Why do not you merge them after a major revision? Please check other figures, e.g., Figs. 3-13. 3. Tables 1-3 seem very similar. Instead
of the original data, might you sum up any common regularity from the three tables? Please check other tables. Answer to referee comment: 1. We will further analyze the data to get more valuable conclusions and expand the discussion of the results. 2. The similar pictures you mentioned should be figures 2-3 and figures 4-13. Similar images will be collated and combined to highlight the differences between the two. 3. The data in tables 1-2 are somewhat similar because they are all taken from northwest China, where debris flow characteristics are very similar. This is also why the data in tables 3-4 are somewhat similar. It is precisely because of the distinct regional characteristics of debris flow in China that we divide the study area into southwest region and northwest region. Thank you for the kind advice.

---

## Author Comment (AC2) · 16 Feb 2020

I am very grateful to your comments for the manuscript. According with your advice, we amended the relevant part in manuscript. Some of your questions were answered below. Major comments: 1.Authors should introduce the description of the occurrence mechanism of the analyzed debris flows: are they run off generated debris flows? Are they channelized debris flows? References about them should be also inserted. 2.Channelized debris flows have a very high impact on the environment because the grow volumetrically along routing with values also up to 100000 m3. The examined

factors should depend on such analysis. 3.About the factors, authors should better distinguish between longitudinal slope and main ditch slope. What do the authors mean for ditch? 4.About factors the presence of sediment source area can play a significant role and maybe the hourly precipitation is better than the daily precipitations. 5.Some sentences are unclear, with no apparent meaning and out of the context. 6.The discussion of results seems poor. The writer suggests the authors to re-write the paper, better explaining the phenomenon, linking the factors to the physics of debris flow occurrence and widening the discussion of results. Answer to referee comment: 1.We are extremely grateful to you for pointing out this problem. I have read the article you mentioned in detail, and I will add relevant content and relevant references. 2. As a result of your suggestion, I have found the shortcomings in my current work. I will improve my scientific research level in accordance with your suggestion in the future work and make more achievements. 3. Thanks for your question, I have checked the corresponding content in the article again and found my own major mistake. The vertical slope should be the ratio of longitudinal slope, that is, the ratio of the difference between the elevation of the gully source and the gully mouth of the debris flow to the length of the main gully. Ditch means main channel of debris flow. 4. The presence of sediment source area you mentioned is really meaningful, but the data collected is not enough to support this analysis. In addition, due to the sudden outbreak of debris flow, it is difficult to monitor the specific time of debris flow eruption in each region, so daily rainfall is selected. 5.We regret there were problems with the English. The paper has been carefully revised by a professional language editing service to improve the grammar and readability. 6. We will analyze the data further to get more valuable conclusions. Thank you for the kind advice.

---

## Author Comment (AC4) · 16 Feb 2020

I am very grateful to your comments for the manuscript. According with your advice, we amended the relevant part in manuscript. Some of your questions were answered below. Major comments: 1. As noted in abstract: "The occurrence of debris flow is often affected by hydro-geological and geological conditions, including basin area, main ditch length, relative height difference, slope, bed bending coefficient, daily maximum rainfall..." However, only six parameters have been selected. 2. All six parameters can classified into two groups, geomorphologic and rainfall, and parts are too similar.

Nevertheless, what are the effects of other factors, e.g. lithological and structural conditions, vegetation, human activity etc.... 3. Potential advantages of meta-analyses are clear, however, they also have the potential to mislead seriously, e.g. study designs, within-study biases, variation across studies, and reporting biases etc... Moreover, like any tool, statistical methods can be misused. The phenomena of selective outcome reporting and publication bias are likely occurred in this manuscript. 4. The tables and figures are not well prepared, and may able to merged and simplified 5. No major conclusion. Answer to referee comment: 1. As for your question, lines 76-78 of the article explain it. The reason for choosing them is that these six factors have obvious digital characteristics and quantifiable characteristics. 2. The lithological and structural conditions you mentioned are indeed of great significance. However, since there are obvious differences between northwest and southwest China in these two aspects, researchers always take these two factors into consideration when evaluating this region, so the statistical evaluation of these two factors is of little significance. Due to the lack of obvious numerical and quantifiable characteristics of vegetation and human activities, selecting them for analysis may lead to seriously misleading. results. 3. As a result of your suggestion, I have found the shortcomings in my current work. I will improve my scientific research level and make more achievements in the future work according to your suggestion! 4. We will merge and simplify the figures and further organize the tables. 5. We will further analyze the data to get more valuable conclusions and expand the discussion of the results. Thank you for the kind advice.